# Adenovirus protein VII binds the A-box of HMGB1 to repress interferon responses

**Edward A. Arnold[1], Robin J. Kaai[2], Katie Leung[1], Mia R. Brinkley[3], Laurel E. Kelnhofer-Millevolte[4], Monica S. Guo[1], Daphne C. Avgousti** [1,3]*

**1** Department of Microbiology, University of Washington, Seattle, Washington, United States of America, **2** Molecular & Cellular Biology, Graduate Program, University of Washington, Seattle, Washington, United States of America, **3** Human Biology Division, Fred Hutchinson Cancer Center, Seattle, Washington, United States of America, **4** Medical Scientist Training Program, University of Washington, Seattle, Washington, United States of America

* avgousti@fredhutch.org

## Abstract

Viruses hijack host proteins to promote infection and dampen host defenses. Adenovirus encodes the multifunctional protein VII that serves both to compact viral genomes inside the virion and disrupt host chromatin. Protein VII binds the abundant nuclear protein high mobility group box 1 (HMGB1) and sequesters HMGB1 in chromatin. HMGB1 is an abundant host nuclear protein that can also be released from infected cells as an alarmin to amplify inflammatory responses. By sequestering HMGB1, protein VII prevents its release, thus inhibiting downstream inflammatory signaling. However, the consequences of this chromatin sequestration on host transcription are unknown. Here, we employ bacterial two-hybrid interaction assays and human cell culture to interrogate the mechanism of the protein VII-HMGB1 interaction. HMGB1 contains two DNA binding domains, the A- and B-boxes, that bend DNA to promote transcription factor binding while the C-terminal tail regulates this interaction. We demonstrate that protein VII interacts directly with the A-box of HMGB1, an interaction that is inhibited by the HMGB1 C-terminal tail. By cellular fractionation, we show that protein VII renders A-box containing constructs insoluble, thereby acting to prevent their release from cells. This sequestration is not dependent on HMGB1's ability to bind DNA but does require post-translational modifications on protein VII. Importantly, we demonstrate that protein VII inhibits expression of interferon β, in an HMGB1-dependent manner, but does not affect transcription of downstream interferon-stimulated genes. Together, our results demonstrate that protein VII specifically harnesses HMGB1 through its A-box domain to depress the innate immune response and promote infection.

## Author summary

Adenoviruses can infect many cell types and cause diseases ranging from pink eye to respiratory infections. To successfully takeover the cell, adenoviruses must pinpoint and hijack host processes to create an environment ideal for producing viral progeny. Adenovirus encodes a small protein, protein VII, that is critical to viral takeover of the host cell

---

**Data Availability Statement:** The authors confirm that all data underlying the findings are fully available without restriction. All relevant data are

within the paper and its Supporting Information files.

**Funding:** This study was supported by NIH funding to E.A.A. (T32AI083203), M.S.G. (R00GM134153), and D.C.A. (R35GM133441), and startup funds from the University of Washington (M.S.G., https://microbiology.washington.edu/) and Fred Hutchinson Cancer Center (D.C.A., https://www.fredhutch.org/en.html). The funders had no role in study design, data collection and analysis, decision to publish, or preparation of the manuscript.

**Competing interests:** The authors have declared that no competing interests exist.

both to disrupt normal cellular function and prevent downstream inflammatory responses. Protein VII binds to the host chromatin where it disrupts the cell cycle and alters the shape of the nucleus, with catastrophic consequences for the cell. In this study, we used a multifaceted approach to show how and where protein VII interacts with a specific and significant inflammatory mediator protein called HMGB1. We further demonstrated the outcome of this interaction on transcription of interferon itself, a key player in the host immune arsenal. Our study provides important biochemical insight into viral strategies to evade immune responses.

## Introduction

Nuclear-replicating viruses interact with and hijack host chromatin to promote viral replication, evade the immune system, and create viral progeny. Adenovirus is a double-stranded DNA virus that manipulates the nuclear environment and host chromatin to promote a productive infection [1,2]. Adenoviruses encode an essential core viral protein, known as protein VII, that packages with the viral genome inside virions [3–5]. Protein VII is initially translated as a precursor protein, pre or pVII, which is cleaved by a viral protease to remove the N-terminal 24 amino acids and produce the packaged mature protein VII [6–8], herein referred to as protein VII. Protein VII is delivered with the viral genome into the nucleus and is released upon initiation of viral transcription [9–11]. Late during infection, high levels of protein VII are produced, the majority of which localize to and distort host chromatin [12]. Protein VII is also post-translationally modified at five sites (K2-acetyl, K24-acetyl, T32-phospho, T50-phospho, S159-phospho) that are responsible for localization of protein VII to host chromatin [12,13]. When all five are mutated to alanine, abrogating modification, protein VII localizes to the nucleolus instead of chromatin [12]. Virus lacking protein VII can be generated, but these virions cannot establish infection and are markedly less stable [14,15], indicating that protein VII is dispensable for virion formation but not infection.

During infection, protein VII interacts with and sequesters in chromatin the host alarmin high mobility group box 1 (HMGB1) [12,16]. HMGB1 is a multifunctional protein that acts both extracellularly as an alarmin and within the cell as a transcription co-factor [17]. In the nucleus, HMGB1 is one of the most abundant proteins that transiently binds to DNA in a non-sequence specific manner, subsequently inducing DNA bending [18,19]. HMGB1 consists of three domains, an A-box and B-box that function to bind DNA [20], and a highly charged acidic C-terminal tail. The A-box and B-box domains have bulky hydrophobic residues that insert into the minor groove of DNA, as well as basic residues that bind the phosphodiester backbone, allowing HMGB1 to bend DNA to extreme angles (reviewed in [21]). The C-terminal tail, a 30 amino acid stretch of glutamic and aspartic acid residues, interacts with the A- and B-box domains to regulate their binding to DNA or other proteins [22–25].

Upon cellular stress, HMGB1 is shuttled from the nucleus to the cytoplasm and is rapidly released from the cell [26]. HMGB1 is therefore considered a typical damage-associated molecular pattern, or DAMP, that induces migration of immune cells and pro-inflammatory cytokines [27]. Extracellular HMGB1 has two characterized cellular receptors, the receptor for advanced glycation endproducts (RAGE) and toll like receptor 4 (TLR4), and is predicted to bind several other host cell receptors as well [28–31]. Interestingly, HMGB1 has also been characterized as a nucleic acid sensor in the cytoplasm where it facilitates innate immune proteins like cyclic GMP–AMP synthase (cGAS) to recognize pathogen-associated molecular patterns (PAMPs), including exogenous RNAs and DNAs [32,33]. The downstream outcome of

these pathways is expression of type-I interferons (IFNs), which are released in turn and lead to the expression of numerous interferon-stimulated genes (ISGs) that have multiple antiviral functions [34–37]. While HMGB1 functions to stimulate and amplify inflammatory responses, it is unknown whether HMGB1's activity as a transcriptional co-factor may also directly impact the transcription of interferon or ISGs. Thus, protein VII preventing HMGB1 release may have multipronged effects on immune responses.

In this study, we demonstrate the mechanism of the protein VII-HMGB1 interaction and the consequence to transcription of immune response genes on the host genome. We show that adenovirus protein VII directly interacts with the A-box of HMGB1, causing HMGB1 to become insoluble and unable to be released from the cell. We further show that the protein VII-HMGB1 interaction is not dependent on HMGB1's association with DNA, nor is it dependent on post translational modifications (PTMs) on protein VII, although protein VII PTMs are required to render HMGB1 insoluble. Finally, we demonstrate that protein VII represses the immune response to infection by dampening transcription of interferon beta (IFNβ) in an HMGB1-dependent manner. This study is the first to demonstrate the mechanism of protein VII hijacking of HMGB1, thereby uncovering a key aspect of immune evasion by adenoviruses.

## Results

### Protein VII directly interacts with HMGB1 by bacterial two-hybrid analysis

To identify which domains of HMGB1 mediate physical interaction with protein VII, we performed a bacterial two-hybrid analysis (B2H) based on the reconstitution of the split *Bordetella pertussis* adenylate cyclase [38]. Briefly, "bait" and "prey" proteins were fused to either the T18 or T25 fragment of adenylate cyclase and co-expressed in *Escherichia coli* (Fig 1A). Protein-protein interaction between bait and prey reconstitutes adenylate cyclase, drives synthesis of cyclic-AMP, and leads to a red colony coloring on MacConkey agar plates (Fig 1A). This B2H system has previously been employed to quantitatively interrogate protein-protein interaction of various bacterial, fungal, and mammalian proteins [38–41]. We first sought to replicate our previous finding that protein VII interacts with HMGB1 [12] using C-terminal fusions of protein VII and HMGB1 to T25 and T18, respectively (T25-protein VII and T18-HMGB1, Fig 1B). We found that T25-protein VII and T18-HMGB1 directly interact (Fig 1C), validating our B2H assay system.

There are three domains of HMGB1: the DNA binding A-box (amino acids 9–79) and B-box (amino acids 89–162), and the acidic C-terminal tail (amino acids 186–215) (Fig 1B). To determine the domain(s) responsible for binding protein VII, we generated several constructs of HMGB1 as follows: only the A-box, the A- and B-boxes together, only the B-box, and the B-box together with the C-terminal tail alone (abbreviated as shown in Fig 1B). We found that loss of the A-box completely abrogated HMGB1 interaction with protein VII, while loss of the C-terminal tail increased interaction (Fig 1C). Next, we asked if either the A-box or B-box was sufficient for protein VII binding and found that protein VII interacted with the A-box and did not interact with the B-box (Figs 1C and S1A). As a control, we repeated the B2H assay with all other possible orientations of protein VII with HMGB1, truncation mutants of the A-box, and A-box alone (S1B Fig, see S1 Table). Although interaction between full-length HMGB1 and protein VII was not observed in any of these alternative orientations, we captured a strong interaction between protein VII and the HMGB1 A-box in all but one orientation tested (S1B Fig, see S1 Table). Together these data indicate that protein VII directly binds to the HMGB1 A-box.

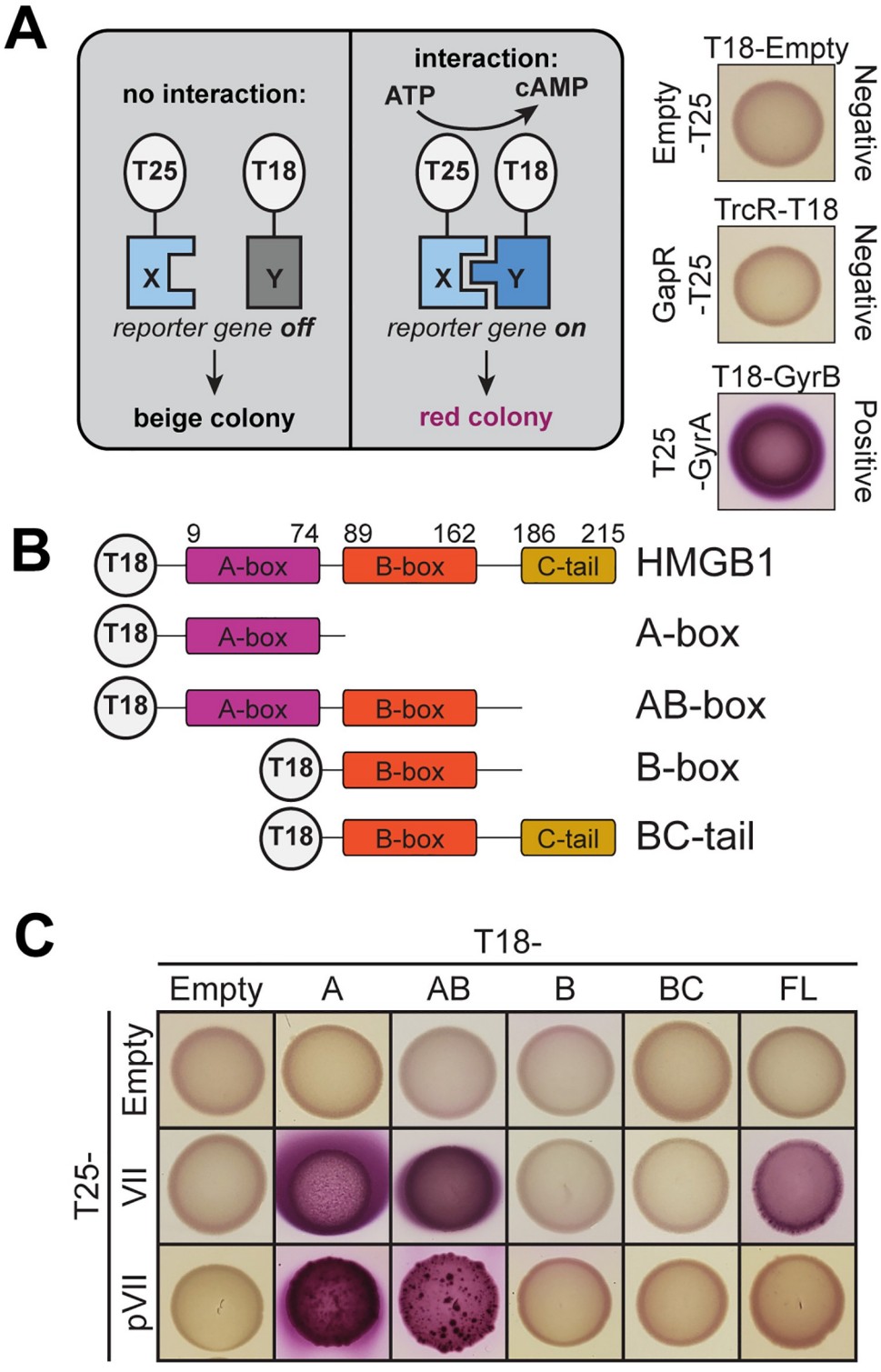

**Fig 1. Protein VII directly binds HMGB1.** *A*. *Left*: Schematic of the bacterial two-hybrid (B2H) assay. *Right*: Empty vectors or no interaction between the bait and prey proteins results in beige colony (top and middle) while a direct interaction results in pink colony (bottom). B. Diagram of different HMGB1 constructs used for B2H assay. C. B2H results of interactions between HMGB1 constructs as indicated above with N-terminal T18. Protein VII constructs are indicated with N-terminal T25. Protein VII strongly binds A-box containing constructs of HMGB1 while pVII only binds A-box and AB-box constructs.

Protein VII is synthesized as a precursor form (pVII) with the precursor portion subsequently cleaved during the late stages of infection during virion maturation [6–8]. We assayed binding of pVII to HMGB1 and discovered that pVII did not interact with full-length HMGB1 due to inhibition from the C-terminal tail, as pVII was able to interact with the A-box and with the AB-box lacking this inhibitory tail (Fig 1C, bottom row).

Lastly, we tested the ability of HMGB1 and protein VII to self-dimerize. Human HMGB1 dimerizes in the presence of reactive oxygen species [42,43], while other HMGB1 homologs, such as human mitochondrial transcription factor A (mtTFA) and HMGa from maize, also form dimers [44,45]. Similarly, protein VII is found as a multimer when packaged with viral genomes, though the stoichiometry of these multimers is unknown [46,47]. We found that full-length HMGB1 did not self-interact, but the A-box did dimerize if C-terminally tagged A-box was paired with N-terminally tagged A-box (T18-Abox with Abox-T25, S1C Fig, see S1 Table). These results suggest that HMGB1 dimerization may only occur when A-boxes are properly oriented and that steric effects or other interactions from the B-box and C-terminal tail may inhibit dimerization. We additionally found that protein VII self-association is extremely weak but is enhanced when the sites of post-translational modification are altered (S1D Fig, see S1 Table), suggesting that the PTM residues may be integral to protein VII self-interaction.

## HMGB1 is co-localized with protein VII during adenovirus infection

Given the direct interaction between protein VII and HMGB1, we next hypothesized that protein VII binds specifically to the A-box of HMGB1 within the nucleus during infection. To test our hypothesis, we generated cell lines expressing different constructs of HMGB1 with a GFP tag and visualized localization in the presence or absence of adenovirus infection (Figs 2A and S2). Given that HMGB1 and its homologs can self-associate [42–45] and our findings above show that HMGB1 can self-dimerize through the A-box, we used A549 cells devoid of endogenous HMGB1 through CRISPR-Cas9 knock-out [13] and introduced our constructs into this null background. Thus, we were able to interrogate protein VII's interaction with our HMGB1 constructs without the complication of dimerization with endogenous HMGB1. We first validated that all cell lines expressed HMGB1 constructs to comparable levels both with each other and endogenous HMGB1 by western blot (Fig 2B). We then visualized the localization pattern of the HMGB1 constructs by immunofluorescence microscopy and found that the A-box, B-box, and BC-tail were diffusely nuclear while the AB-box was somewhat nucleolar (Fig 2C, mock). Of note, the A-box, while mostly nuclear, had an observable cytoplasmic signal. This is likely due to lack of the second reported NLS site on HMGB1 in this construct (residues 179–185) [48]. Upon infection with human adenovirus type 5 (Ad5), we observed that the HMGB1 constructs had three different localization patterns: a diffusely nuclear pattern similar to that of mock infection (Fig 2C, triangle), a ring-like pattern, referred to as pattern I (Fig 2C, circle), or a punctate pattern, referred to as pattern II (Fig 2C, star). We quantified the different patterns seen during infection for each cell line and plotted the frequency of the pattern as a percentage (Fig 2D). We observed that all constructs containing an A-box had noticeable shift away from diffuse to patterns I and II. Full length HMGB1 had an even distribution of all three patterns whereas the A-box and AB-box constructs both favored pattern I. Interestingly, the B-box construct also had a observable shift from diffuse to pattern II. Both the BC-tail and GFP cells remained diffusely nuclear, suggesting that these constructs are not being mislocalized during infection.

To investigate whether the changes in pattern are due to differences in binding chromatin, we extracted soluble proteins prior to fixation (termed pre-extraction) and then visualized the

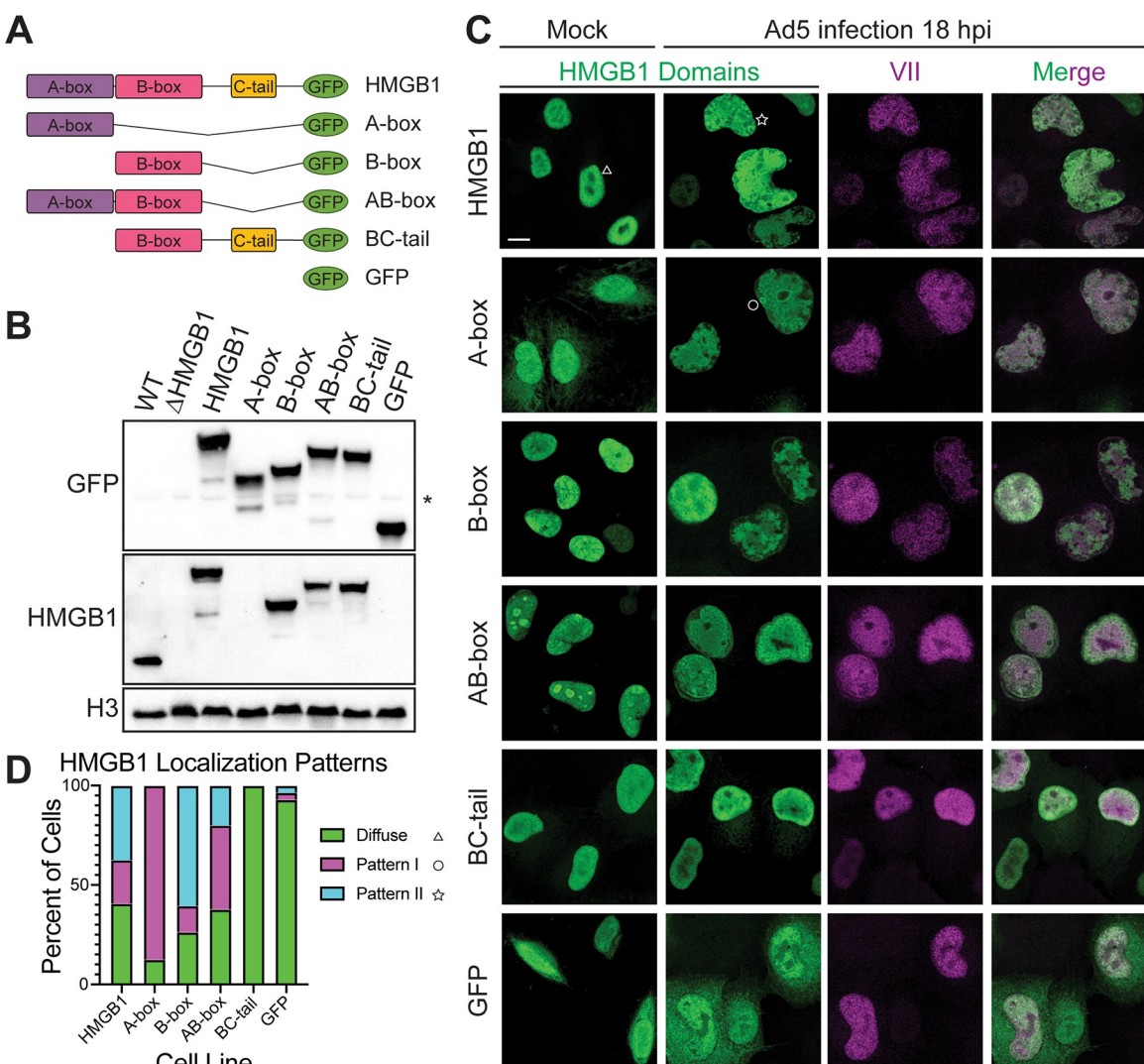

**Fig 2. Protein VII mislocalizes A-box containing constructs of HMGB1 during infection.** A. Schematic of HMGB1 constructs expressed in HMGB1-deleted A549 cells. B. Representative western blot showing expression from cell lines generated as in (A) indicating similar expression levels to endogenous HMGB1. *Top*: GFP, *middle*: HMGB1, *bottom*: H3 loading control. *Note*: HMGB1 antibody binds the B-box, thus no signal is detected for A-box cell line, * denotes non-specific band. C. Immunofluorescence images of cell lines as in (A) showing HMGB1 domains in green and protein VII in magenta. Left column is mock-infected cells showing HMGB1 construct distribution and right three columns are Ad5 infected, MOI of 10, at 18 hours post infection (hpi). Scale bar is 10 μm. D. Table summarizing localization changes of HMGB1 constructs during infection as presented in (C). Frequency of HMGB1 localization patterns observed for each cell line depicted as percentage of total cells. N>20 cells for each cell line.

patterns by immunofluorescence microscopy as above. Furthermore, we used a virus with HA-tagged protein VII (Ad5 VII-HA) to allow for better resolution of protein VII with a commercial antibody and focused on constructs with or without the A-box. When we infected our HMGB1, AB-box, and BC-tail cells with Ad5 VII-HA and fixed the cells under the previously used conditions, as expected we observed similar patterns to our Ad5 WT virus (compare Fig 2C with Fig 3A). When we pre-extracted mock infected cells, soluble HMGB1 leaked out of the nuclei in all conditions such that HMGB1 is no longer visible upon imaging (Fig 3B). Imaging Ad5 VII-HA infected cells after pre-extraction revealed full-length HMGB1 and the AB-box construct remained nuclear and the signal fully overlapped with that of protein VII

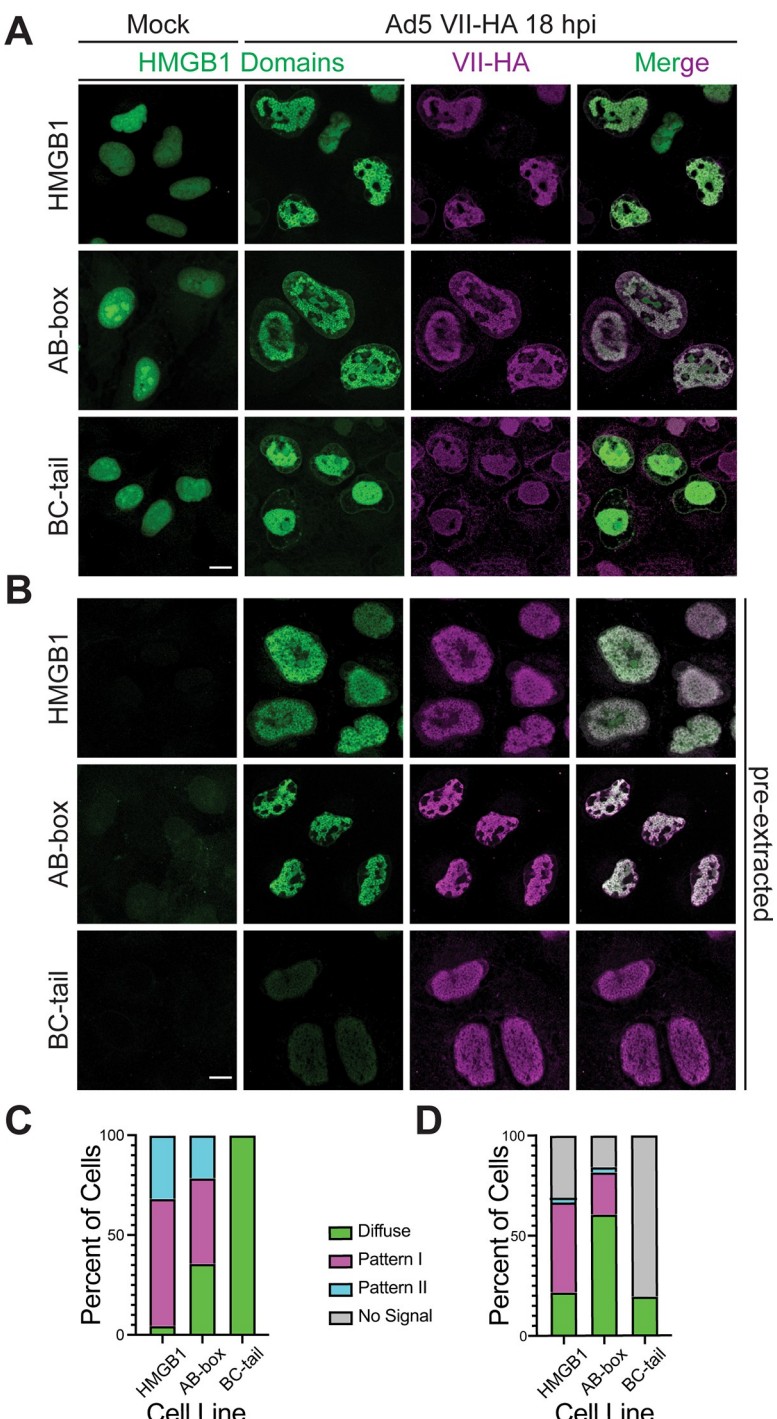

**Fig 3. Protein VII retains A-box containing constructs of HMGB1 in the chromatin.** A. Immunofluorescence images of HMGB1, AB-box and BC-tail cell lines, showing HMGB1 domains in green and protein VII in magenta. Left column is mock-infected cells showing HMGB1 construct distribution and right three columns are Ad5 VII-HA infected, MOI of 10, at 18 hours post infection (hpi). Scale bar is 10 μm. B. Same as in (A), but cells were pre-extracted before fixation, staining, and imaging. C. Frequency of HMGB1 localization patterns observed for each cell line depicted as percentage of total cells during Ad5 VII-HA infection. Total N>30. D. Same as in C, but cells were under pre-extraction conditions. N>30 for each cell line.

(Fig 3B). In contrast, the BC-tail construct was extracted along with other soluble proteins, resulting in little to no signal upon imaging (Fig 3B). We quantified these localization patterns (Fig 3C and 3D) and conclude that the A-box is required for protein VII to render HMGB1 bound to the chromatin during infection.

## Protein VII is sufficient to render HMGB1 insoluble

The release of HMGB1 from cells is a key step in inflammatory signaling. Therefore, we hypothesized that protein VII may prevent HMGB1 release by immobilization in chromatin through direct interaction. Because HMGB1 release is critical for extracellular signaling, we chose to use cellular fractionation to measure HMGB1 solubility as a proxy for HMGB1 release. We used simple fractionation by mild detergent to separate cells (termed whole cell lysate or WCL) into either soluble or insoluble fractions (Fig 4A). Soluble nuclear and cytoplasmic proteins are found in the soluble fraction (termed Sol), while insoluble proteins, made up of chromatin-bound proteins and histones, are found in the insoluble fraction (termed Ins) (Fig 4B). We used this method to assay the solubility of several proteins during infection and found vinculin, a large cytoskeletal protein, to be soluble, histone H3 to be insoluble, and

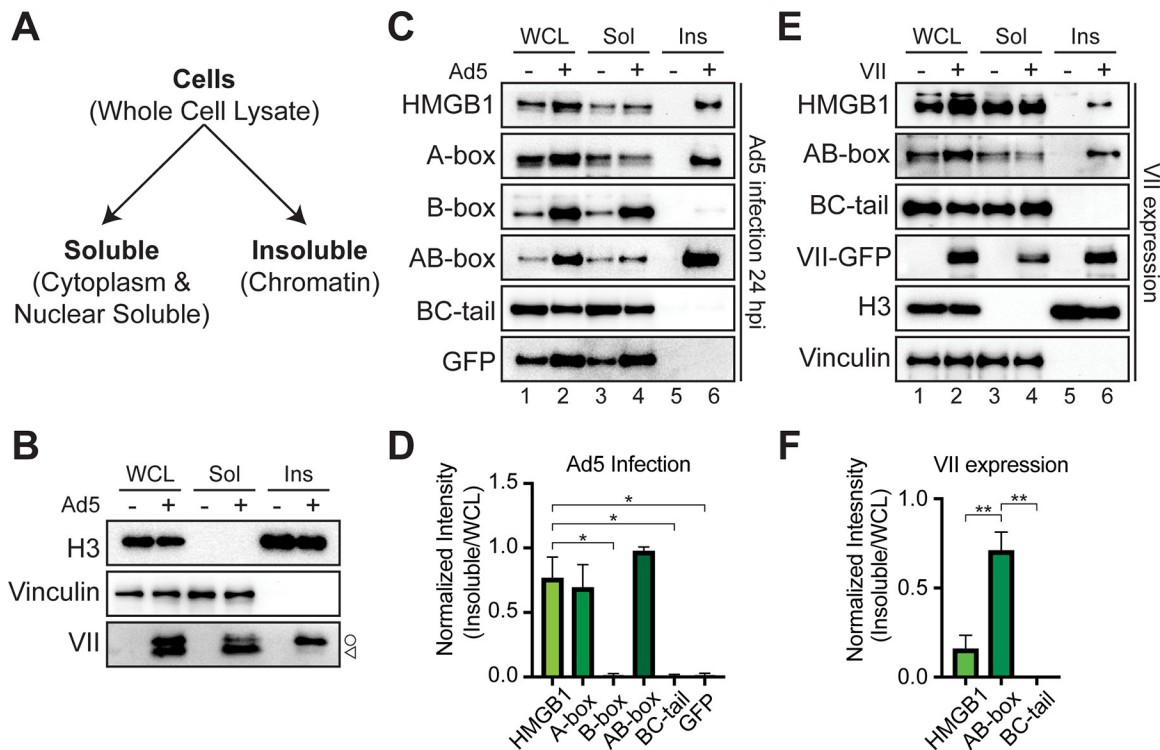

**Fig 4. Protein VII causes A-box containing constructs of HMGB1 to be insoluble.** A. Schematic of fractionation assay. B. Western blot of fractionated cells showing representative blots of three biological replicates. Whole cell lysate (WCL) in left two lanes, soluble (Sol) fraction in middle two lanes, and insoluble (Ins) fraction in right two lanes with or without Ad5 infections as indicated by +. H3 indicates insoluble chromatin while vinculin is cytoplasmic and soluble. C denotes pVII and < is mature protein VII. C. Representative western blots of fractionated Ad5 infected cells at 24 hpi of cell lines as indicated showing predominantly A-box containing constructs of HMGB1 are rendered insoluble during infection. Lanes are numbered 1–6. 1: WCL mock, 2: WCL infected, 3: soluble mock, 4: soluble infected, 5: insoluble mock, 6: insoluble infected. D. Densitometry of western blots in (C) of HMGB1 in the insoluble fraction of infected cells normalized to HMGB1 in the WCL fraction. Error bar represents SD of three biological replicates, * is p<0.05 by Welch's t-test. E. Representative western blots upon ectopic protein VII expression of cell lines as indicated showing that protein VII is sufficient to render A-box containing constructs insoluble. Lanes numbered 1–6, same as (C) with expression of protein VII instead of infection. F. Same as (D) for western blots from (E). Error bar represents SD of three biological replicates, ** is p<0.01 by Welch's t-test.

protein VII in both soluble and insoluble fractions (Figs 4 and S3), consistent with previous findings [12,49]. Interestingly, we were able to resolve the two bands of protein VII and pVII and found that pVII is predominantly insoluble while the mature form is more soluble, suggesting a separation of properties that may be indicative of function. We next investigated the solubility of GFP-tagged HMGB1 constructs from the cell lines described above using this fractionation assay. We found that in the absence of infection, all HMGB1 constructs are soluble (Fig 4C, compare lanes 3 and 5). In contrast, during infection we find that full length HMGB1, A-box only, and AB-box, that is constructs containing the A-box, are insoluble whereas the B-box alone and BC-tail constructs are soluble (Fig 4C, lane 6). There is a slight band present for the B-box alone in the insoluble fraction during infection, suggesting that there may be weak retention of the B-box in chromatin. When the C-terminal tail is added to the B-box, there is no observable signal in the insoluble fraction. Because all HMGB1 constructs have the same GFP tag, we were able to directly compare the levels in the insoluble fraction by densitometry of the western blots normalized to the infected WCL. We found that there is significantly more HMGB1 retained in the insoluble fraction of the full length, A-box, and AB-box cells compared to the negative GFP control (Fig 4D). Importantly, we examined viral protein levels in our cell lines and observed that they were unaffected by the various HMGB1 constructs (S4 Fig), suggesting that the impact of the protein VII-HMGB1 interaction is likely beyond a single infected cell. This notion is consistent with insolubility as a proxy for HMGB1 release affecting systemic inflammatory responses to infection.

To determine whether protein VII was sufficient for the change in solubility of the HMGB1 constructs, we expressed protein VII alone and investigated HMGB1 solubility. To do this, we transduced cells with an E1-deleted replication incompetent recombinant adenovirus (rAd) that expresses only protein VII-GFP and assayed for solubility. We found that indeed protein VII is sufficient to render HMGB1 insoluble, consistent with previous reports [12]. Further, we found that only A-box containing constructs were rendered insoluble by protein VII (Fig 4E), consistent with our findings above that protein VII directly interacts with the A-box. Interestingly, the retention of full length HMGB1 in the insoluble fraction is much weaker than that of the AB-box, further supporting the idea that the C-terminal tail may inhibit the interaction between protein VII and the A-box. Consistent with this interpretation, our densitometry analysis confirmed that there was significantly more insoluble AB-box than HMGB1 or BC-tail (Fig 4F). Taken together, these results indicate that during infection protein VII binds the A-box of HMGB1, rendering it insoluble and sequestered in chromatin.

## Protein VII interacts with, mislocalizes, and sequesters HMGB1 in chromatin independent of HMGB1's interaction with DNA

Since HMGB1 can directly bind DNA [50], we next asked whether the interaction between protein VII and HMGB1 is dependent on this DNA binding. To test this, we changed the three hydrophobic residues in HMGB1 that are responsible for the DNA interaction to alanine: F38A, F103A and I122A (mutHMGB1, Fig 5A), and investigated the protein VII interaction by B2H assay. F38 is found in the A-box while F103 and I122 are in the B-box (Fig 5A). We tested protein VII interaction with mutated HMGB1 in the A-box only, the AB-box and full-length HMGB1 (FL) and found that these mutations did not eliminate interaction with protein VII (Fig 5B). We note that although the protein VII with AB-box and FL B2H colonies are red, indicating a positive interaction, the colonies exhibit a speckled pattern (compare to A-box or to Fig 1C), indicating that there may be a decrease in interaction efficiency. To further exclude the possibility that our B2H assays captured an indirect, DNA-mediated interaction between HMGB1 and protein VII, we validated that two non-specific DNA-binding proteins do not

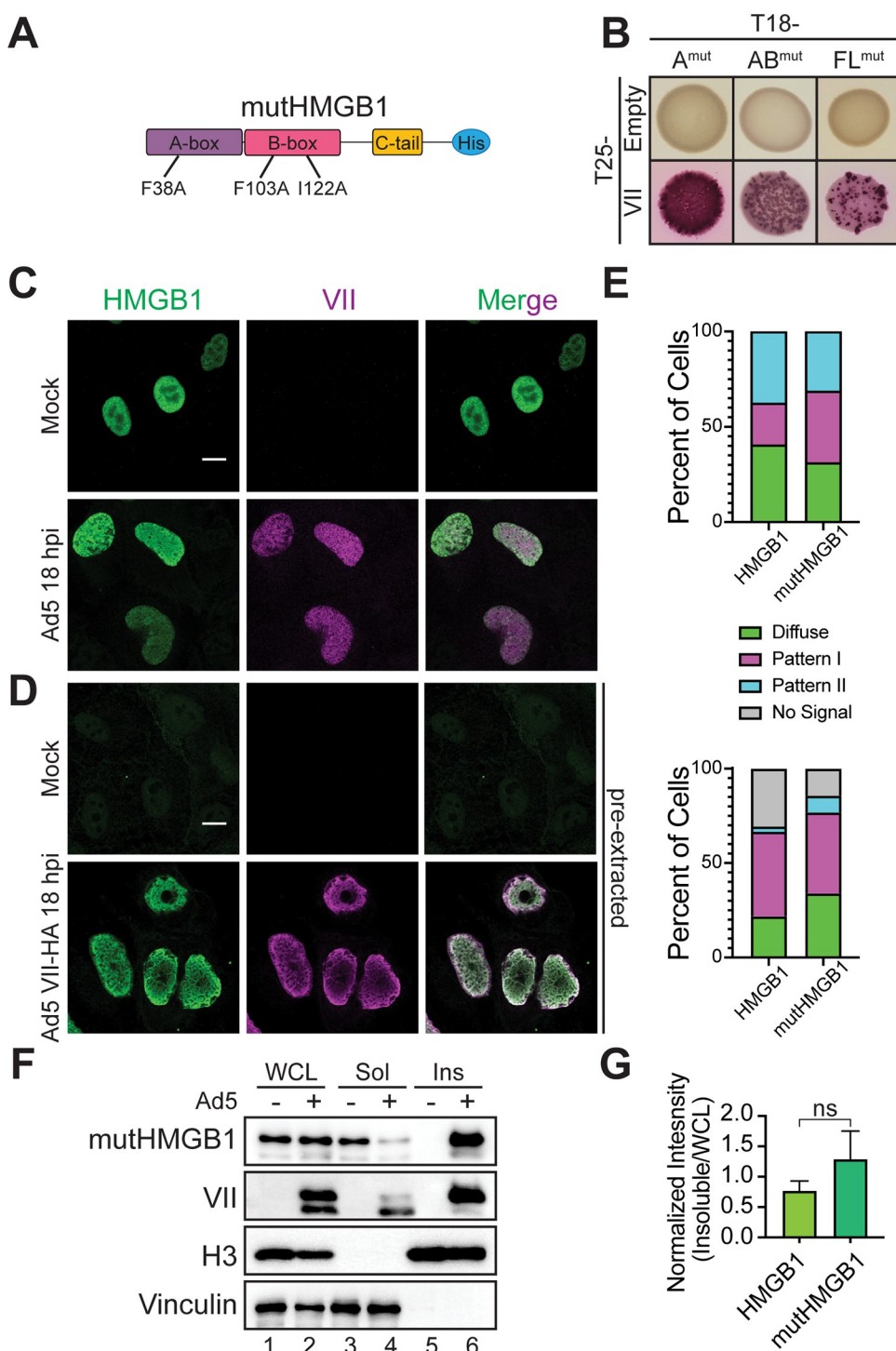

**Fig 5. Protein VII's interaction with HMGB1 is not dependent on HMGB1's interaction with DNA.** A. Schematic of mutHMGB1 expressed in A549ΔHMGB1 cells with point mutations as indicated. B. B2H results of interaction between protein VII and mutHMGB1 constructs as indicated. C. Immunofluorescence images of full length mutHMGB1 cells as in (A) showing HMGB1 in green and protein VII in magenta. Top row is mock infected showing mutHMGB1 distribution and bottom row is Ad5 infected, MOI of 10, at 18hpi. Scale bar is 10 μm. D. Same as in (C), but under pre-extraction conditions, and infection is with Ad5 VII-HA. Scale bar is 10 μm. E. Frequency of HMGB1 localization patterns observed for each cell line depicted as percentage of total cells during Ad5 infection for top graph. N>30 for each cell line. Bottom graph is same as top, but for Ad5 VII-HA infection and under pre-extraction conditions. N>30 for each cell line. F. Representative western blots of fractionated Ad5 infected mutHMGB1 cells at

24 hpi showing mutHMGB1 constructs are rendered insoluble in the presence of infection. Lanes as in Fig 3C. G. Densitometry of western blots as in (D) of mutHMGB1 in the insoluble fraction of infected cells normalized to HMGB1 in the WCL fraction. Wild type HMGB1 results from Fig 3E. Error bar represents SD of three biological replicates, * is p<0.05 by Welch's t-test.

interact in the B2H assay (Fig 1A). Therefore, we conclude that these mutations to HMGB1 do not abrogate binding to protein VII.

We next investigated the interaction of mutHMGB1 with protein VII in human cells. We expressed mutHMGB1 in A549ΔHMGB1 cells and infected these cells with Ad5. We observed by immunofluorescence microscopy that mutHMGB1 localization closely resembled that of wild-type (WT) HMGB1: mutHMGB1 was disperse throughout the nucleus in the absence of infection and a mixture of patterns I and II upon Ad5 infection (Fig 5C, compare to Fig 2C). We also performed pre-extraction on Ad5 VII-HA infected mutHMGB1 cells, as in Fig 3. Much like WT HMGB1, during mock infection mutHMGB1 is not visible within the nucleus of pre-extracted cells but remains bound to the chromatin in infected cells (Fig 5C). These results closely recapitulate those of wild-type HMGB1 suggesting that these mutations do not impact protein VII's ability to bring HMGB1 into insoluble chromatin (Fig 5D–5E). Thus, we conclude that protein VII mislocalizes HMGB1 independent of HMGB1 binding DNA.

Next, we examined the solubility of mutHMGB1 during Ad5 infection. We found that mutHMGB1 is strongly retained in the insoluble fraction upon Ad5 infection (Fig 5D, lane 6). When compared to FL HMGB1 by densitometry analysis, mutHMGB1 had slightly increased retention, although the difference was not statistically significant (Fig 5E). Taken together, these results indicate that protein VII interacts with, mislocalizes, and sequesters HMGB1 in chromatin independently of HMGB1 DNA binding.

## Post-translational modifications on protein VII do not affect binding to HMGB1 but are important for sequestering HMGB1 in chromatin

Protein VII contains five post-translational modifications (PTMs) that affect protein VII's localization in the nucleus [12]. When the PTMs on protein VII are present, protein VII localizes to the host chromatin. In contrast, when these sites are mutated to alanine, referred to as VIIΔPTM (Fig 6A), protein VII localizes to the nucleolus [12]. This change in localization is thought to facilitate protein VII distinguishing between the host and viral genomes during infection as the nucleolus is disrupted during late stages of infection, causing many nucleolar proteins to localize with viral replication centers [12,51]. Because these PTMs affect protein VII localization, we next asked how the PTMs on protein VII affect the interaction with HMGB1 and HMGB1 localization in the nucleus. We first examined the interaction by B2H and found that VIIΔPTM still interacted with the A-box alone and the AB-box construct, indicating that the interaction between protein VII and the A-box is not dependent on these sites. However, we found that VIIΔPTM no longer interacted with FL HMGB1 (Fig 6B), suggesting that the C-terminal tail fully inhibited the interaction in a bacterial context. Furthermore, B2H interactions between VIIΔPTM and A-box alone and the AB-construct led to weakly pink and speckled colonies, again indicating a potential decrease in interaction compared to wild-type protein VII. This speckling pattern was also observed when we examined self-association of VIIΔPTM (S1D Fig). Although, protein VII self-association was extremely weak, VIIΔPTM exhibited a stronger interaction suggesting that these sites may be important for multimerization of protein VII. It is important to note that in bacteria, WT protein VII is unlikely to be modified as the enzymes responsible for modification would only be present in human cells,

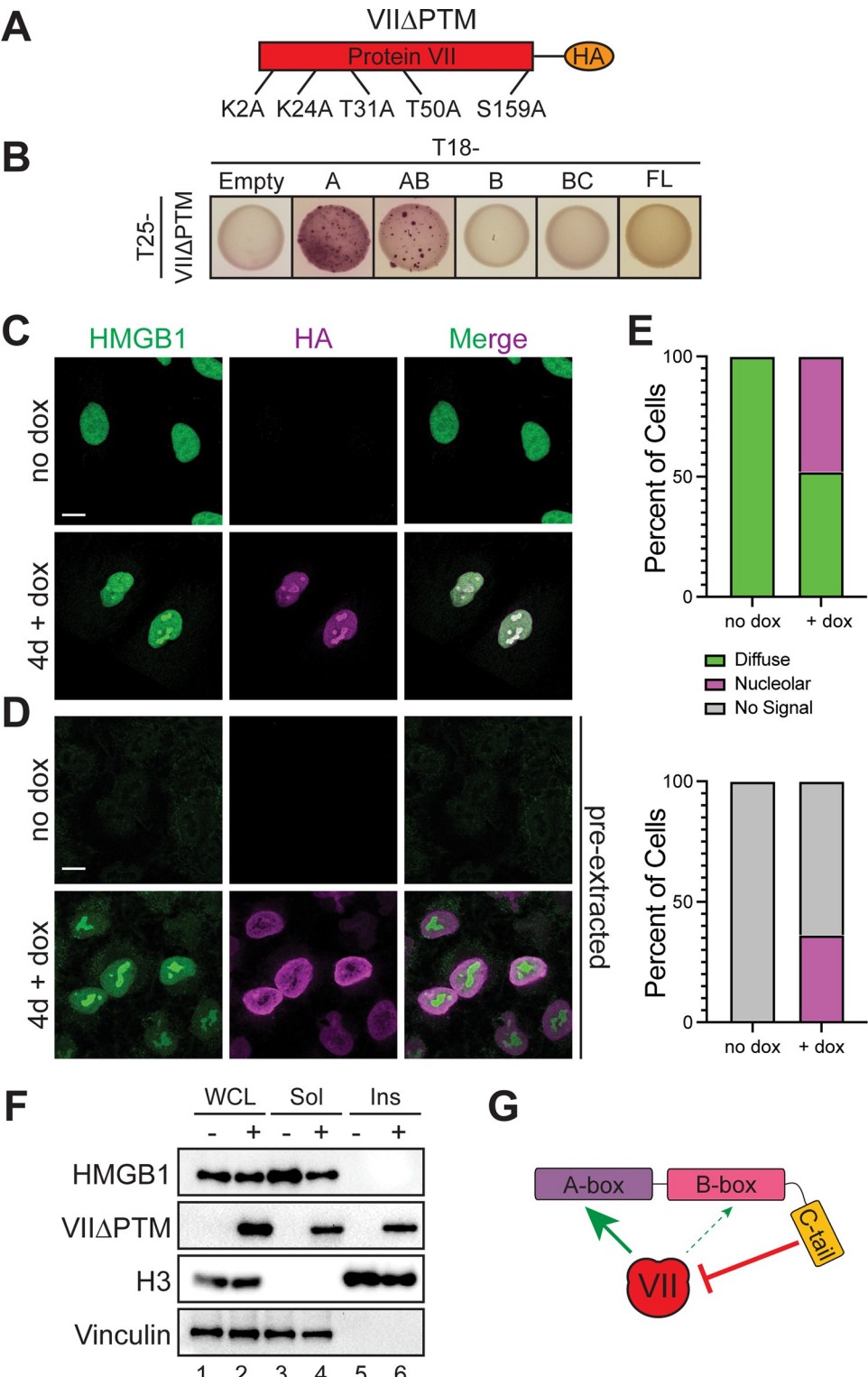

**Fig 6. Post-translational modifications on protein VII are not required for HMGB1 binding but are important to render HMGB1 insoluble.** A. Schematic of point mutations in VIIΔPTM expressed in A549 cells with doxycycline (dox) inducible promoter. B. B2H results of interaction between protein VIIΔPTM and HMGB1 constructs as depicted. C. Immunofluorescence images of A549 cells expressing dox inducible VIIΔPTM, as in (A), with endogenous HMGB1 in green and VIIΔPTM in magenta, showing predominant nucleolar localization of HMGB1 when VIIΔPTM is expressed. Top row is without dox and bottom row is with 4 days of dox treatment. Scale bar is 10 μm. D. Same as in

(C) under pre-extraction conditions. Scale bar is 10 μm. E. Frequency of HMGB1 localization patterns observed for each no dox and + dox conditions, depicted as percentage of total cells for top graph. N>30 for each condition. Bottom graph is for pre-extraction conditions. N>30 for each condition. F. Representative western blots of three biological replicates depicting fractionation of A549 VIIΔPTM cells with and without dox addition. G. Schematic summarizing the interaction of protein VII and HMGB1. Protein VII primarily interacts with the A-box of HMGB1 with some impact on the B-box while the C-terminal tail inhibits the interaction.

therefore, we conclude that these residues are likely important for protein VII multimerization.

We next examined the localization of HMGB1 in the presence of VIIΔPTM in human cells by visualizing endogenous HMGB1 in A549 cells that express HA-tagged VIIΔPTM from a doxycycline (dox) inducible promoter [12]. We observed that without dox, HMGB1 exhibited the characteristic disperse nuclear pattern (Fig 6C). In contrast, upon addition of dox and subsequent induction of protein VIIΔPTM expression, we found that HMGB1 adopted the same pattern as VIIΔPTM and was concentrated in the nucleolus. When pre-extracting the cells prior to fixation, we observed that although the pattern of VIIΔPTM is altered by pre-extraction itself, HMGB1 is strongly localized to the nucleolus when VIIΔPTM is present (Fig 6D). We quantified these patterns as disperse, nucleolar, or no signal (Fig 6E). When we plotted the frequency of these patterns, we found that approximately 50% of the cells have nucleolar HMGB1 when VIIΔPTM is present in either fixation condition. Taken together, these results indicate that VIIΔPTM binds endogenous HMGB1 and causes it to re-localize to the nucleolus.

To investigate how the interaction with VIIΔPTM potentially impacts HMGB1 release, we assayed for solubility by fractionation. Without doxycycline treatment, thus in the absence of protein VII, we found that endogenous HMGB1 was completely soluble, recapitulating HMGB1 solubility in uninfected cells (Fig 6E, lane 3). Interestingly, expression of VIIΔPTM did not cause HMGB1 to become insoluble (Fig 6E, lane 6), unlike WT protein VII. The B2H and microscopy results support a model in which VIIΔPTM binds HMGB1 and the two co-localize to the nucleolus, however, the VIIΔPTM-HMGB1 complex is soluble. This suggests that HMGB1 may not be analogously sequestered in the chromatin. Taken together, these results indicate that while VIIΔPTM can bind the A-box of HMGB1 and mislocalize endogenous HMGB1 to the nucleolus, it cannot cause HMGB1 to be insoluble, suggesting that the PTMs of protein VII may be required to sequester HMGB1 in chromatin.

## Protein VII inhibits interferon signaling by decreasing IFNβ1 expression

Since protein VII immobilizes HMGB1 in chromatin and HMGB1 is implicated as a transcriptional co-factor [52–54], we hypothesized that protein VII may be harnessing this role of HMGB1 to promote immune evasion. Protein VII was originally described as a transcriptional repressor [55], therefore, we hypothesized that protein VII may hijack HMGB1 to repress the innate immune response. To investigate this hypothesis, we used A549 cells that express dox-inducible protein VII [12], similar to the cells used for VIIΔPTM in Fig 5. We induced expression of protein VII, and then treated the cells with interferon to induce the expression of downstream ISGs. We used quantitative PCR to measure the levels of ISGs in the presence or absence of dox and interferon. In the absence of protein VII, we found a robust increase in MX2 and ISG15, but not upstream NF-kB (Fig 7A). When we expressed protein VII, we saw that ISG induction was equally robust, suggesting that protein VII is not affecting the downstream effects of the type-I interferon response (Fig 7A).

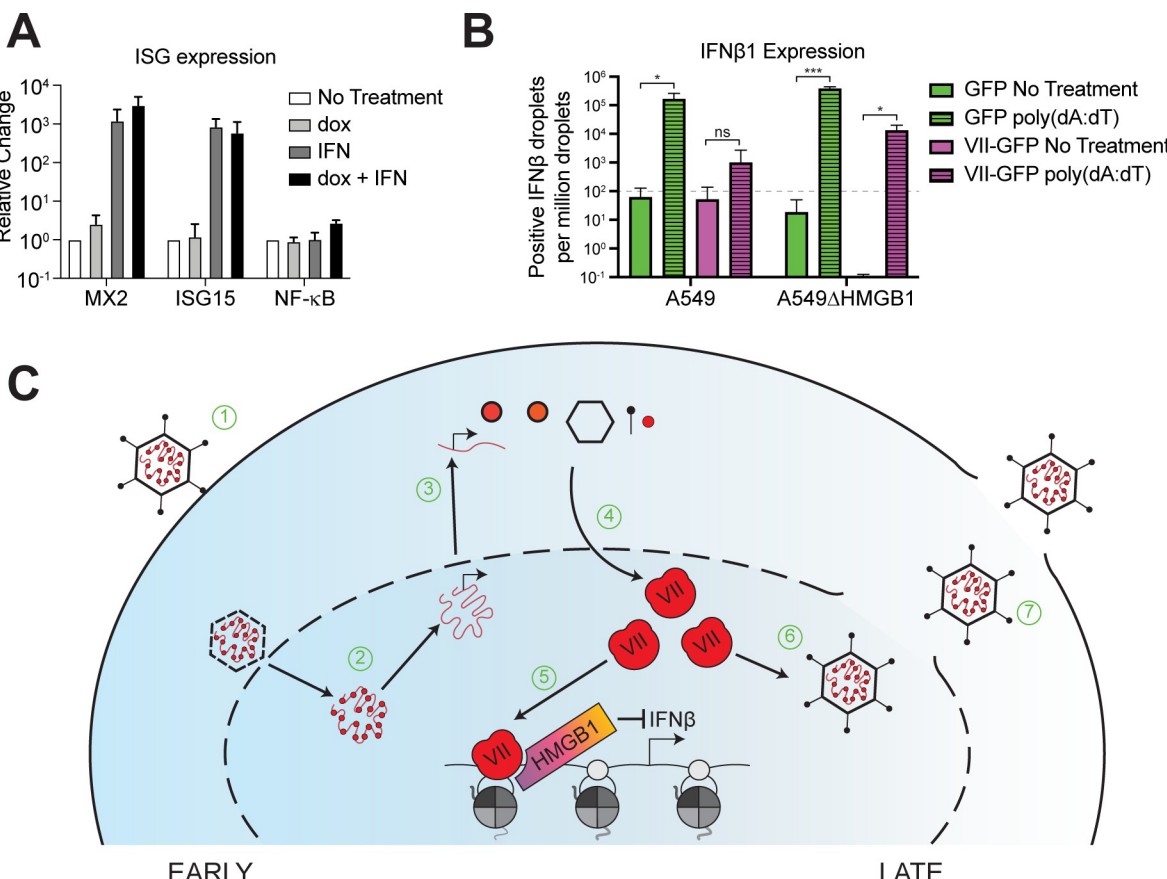

**Fig 7. Protein VII inhibits expression of IFNβ1.** A. Quantification MX2, ISG15 and NF-κB mRNA levels by RT-qPCR in A549 cells with dox inducible protein VII treated with or without human IFNβ as indicated. Error bar represents SD of three biological replicates. B. Quantification of IFNβ mRNA levels by ddPCR in A549 and A549ΔHMGB1 cells with or without protein VII as indicated, with or without poly(dA:dT) treatment as indicated. Dashed line indicates a limit of detection, where anything at or below this line is considered baseline expression of IFNβ. Error bar represents SD of three biological replicates. * is p<0.05, *** is p<0.005, ns is not significant by Student's t-test. C. Schematic detailing adenovirus infection with proposed mechanism of action by protein VII binding HMGB1 for inhibition of IFNβ. 1) Adenovirus enters into the cell and nucleus. 2) Protein VII is removed from the viral genome, histones take its place, and viral transcription begins. 3) Viral transcripts are translated in the cytoplasm. 4) Newly translated protein VII accumulates in the nucleus at later times during infection. 5) Protein VII binds HMGB1, sequesters it on host chromatin, where it inhibits transcription of Ifnβ, thus dampening the type-I interferon response. 6) Protein VII, without HMGB1, is packaged with progeny viral genomes. 7) Viral progeny are released through cell lysis.

Because protein VII strongly binds to chromatin and sequesters HMGB1 [12], we hypothesized that any effect on immune responses may be upstream of ISGs, such as the expression of interferon itself. To test this, we expressed protein VII-GFP or a GFP control (as in Fig 4E) in wildtype (WT) A549 cells and A549ΔHMGB1 cells. We then treated these cells with poly (dA:dT) to mimic exogenous dsDNA and measured IFNβ mRNA levels by droplet-digital PCR (ddPCR) (Fig 7B). In this assay, we normalized IFNβ expression to 18S expression. Since 18S is highly expressed, we normalized IFNβ droplets to every 1 million 18S-positive droplets. While effective, this method does cause relatively low IFNβ droplet amounts (e.g. 1–5) in untreated samples to be overrepresented upon normalization. Thus, in Fig 7B we added a limit of detection line to delineate baseline expression of IFNβ as anything at or below this line. In A549 WT cells without protein VII, we observed a robust increase in IFNβ expression upon stimulation with poly (dA:dT) compared to mock treated (Fig 7B). Upon expression of protein VII, we found a two-log decrease in the levels of IFNβ compared to poly(dA:dT) treated cells

without protein VII. The decrease of IFNβ induction resulting from protein VII expression was not significantly greater than baseline or untreated conditions. These findings strongly indicate that protein VII inhibits expression of IFNβ. In the absence of HMGB1 (A549ΔHMGB1 cells), we found a robust increase in IFNβ expression upon stimulation with poly(dA:dT). While the increase reached significantly higher levels than those of the GFP expressing A549 cells with poly(dA:dT), the magnitude of the difference is small and likely not biologically significant. When the A549ΔHMGB1 cells expressing VII-GFP were treated with poly(dA:dT), we still detected a decrease in IFNβ levels compared to the GFP expressing control conditions, however, to a lesser extent than the decrease in A549 WT cells. Thus, we conclude that protein VII causes a significant reduction in IFNβ expression that is dependent on the presence of HMGB1. Because the loss of HMGB1 did not fully rescue protein VII from dampening IFNβ induction to baseline levels, we predict that other factors such as HMGB2 may also support protein VII's repressive function. Taken together, these results indicate that protein VII binds the A-box of HMGB1 to inhibit interferon signaling by decreasing the expression of IFNβ.

## Discussion

Manipulation of the nuclear environment and the extracellular milieu is critical to the success of nuclear-replicating viruses such as adenovirus. Specifically, adenovirus protein VII prevents the release of HMGB1 and inhibits HMGB1-associated downstream inflammation [12,16]. We hypothesized that protein VII retains HMGB1 in the chromatin not only to prevent its activity as a DAMP, but also to affect transcription; in this case, by inhibiting interferon β expression. In this study, we leveraged bacterial and human culture systems to define the mechanism of protein VII binding to HMGB1 and the consequent effect on host responses.

In our B2H analysis, we demonstrated that both the protein VII precursor, pVII, and mature protein VII directly bind to the A-box or AB-box of HMGB1. In contrast, only mature protein VII binds full-length HMGB1 (Fig 1). The preference of mature protein VII over pVII for HMGB1 is intriguing, as it points to a potential mechanism of the first 24 amino acids in pVII inhibiting the interaction with HMGB1. Interestingly, we also noted by western blot that it was mostly pVII that was found in the insoluble fraction during infection when HMGB1 was also insoluble (Fig 4B), suggesting that in human cells, HMGB1 likely does interact with pVII. One possibility is that pVII is also post-translationally modified in the first 24 amino acids and that these modifications increase affinity for HMGB1. Indeed, we previously reported two modifications in the precursor fragment, S18 phosphorylation and K19 acetylation. These marks identified by mass spectrometry of pVII purified from infected cells may be important for mediating the HMGB1 interaction [12]. These observations indicate that the precursor fragment may be important for modulating the protein VII-HMGB1 interaction during infection.

At late times during infection, the nucleolus becomes disrupted, and nucleolar proteins co-localize with viral replication centers [51]. This is particularly significant in light of the observation that VIIΔPTM is localized to the nucleolus upon ectopic expression because the PTMs may promote distinguishing between viral or host genomes. Interestingly, these PTMs were not identified in purified virus particles [12,56], suggesting that their presence may promote host chromatin binding, while their absence causes localization to the viral genomes and eventual packaging [12,16]. We observed that mutation of the PTM sites also promotes self-dimerization in the B2H assay. While these PTMs are not likely to occur in bacteria, it is possible that the amino acids changed provide a structural function for protein VII that has yet to be determined. Future structural studies will shed light on their relevance to binding DNA and

other factors. Furthermore, virus particles without protein VII are less stable than wild type virions [15]. Thus, because protein VII forms a multimer when compacted with the viral genome inside of virus particles [47], it is possible that these sites promote protein VII multimerization and stabilizes the packaged viral genome.

We further demonstrated that point mutations affecting the DNA binding of HMGB1 did not abrogate the interaction of these two proteins (Fig 5). This does not rule out the potential importance of DNA in the interaction between protein VII and HMGB1, only that it is not required by HMGB1. Protein VII also has high affinity for DNA, but this is likewise not required for the protein VII-HMGB1 interaction as the two proteins bind strongly *in vitro* in the absence of any other factors [12]. In contrast, the C-terminal tail diminished HMGB1 affinity for protein VII in all contexts tested. This includes mature protein VII, pVII, and VIIΔPTM. We were unable to examine any interaction with the C-terminal tail alone as it is highly charged and unstable. Nevertheless, previous studies reported that the C-terminal tail negatively regulates the DNA binding capacity of the A- and B-boxes [23,57,58], consistent with our findings. The HMGB1 C-terminal tail also disrupts the interaction with protein VII, suggesting that the high charge and disordered nature of the C-terminal tail is disruptive to protein-protein or protein-DNA interactions. Interestingly, although HMGB1 is a highly conserved protein from yeast to humans, the yeast homolog, HMO1, does not contain an acidic C-terminus [59], suggesting that this feature evolved later as a means of modulating HMGB1 function.

We found a slight effect of protein VII on solubility and localization of the B-box alone during infection. We observed that there is some B-box in the insoluble fraction upon infection but becomes completely soluble upon addition of the C-terminal tail, similar to full-length HMGB1 in the absence of protein VII. Furthermore, the localization pattern of the B-box was altered during infection though it did not copy the pattern of the A-box containing proteins. Instead, the pattern more resembled that of endogenous HMGB1 upon infection with a protein VII-null virus [16]. This pattern was again completely lost upon addition of the C-terminal tail in the BC-tail construct. This suggests that (1) other viral proteins may affect HMGB1 during infection independently of protein VII and that (2) the presence of the C-terminal tail still diminishes this effect.

In our B2H system, several interactions between mutated protein VII or HMGB1 resulted in a speckled pattern compared to a more uniform pink with WT proteins (Figs 1, 4, 5, and S1), suggesting a weaker interaction, although our B2H assay does not inform on kinetics. Finally, direct interaction was specific to certain orientations of HMGB1 and/or protein VII with the T18 and T25 subunits of adenylate cyclase, suggesting that the physical interaction is stereotyped in specific orientations. Future structural and kinetic studies will be needed to define the spatial dynamics of the protein VII-HMGB1 interaction. Together, these observations indicate that the interaction between protein VII and HMGB1 is robust but can be modulated by the C-terminal tail and post-translational modifications. Our data support a model in which VII interacts with the A-box, and very slightly with the B-box, while the C-terminal tail inhibits this interaction (Fig 5E). The process of HMGB1 release from the cell is heavily reliant on modification of HMGB1 and the oxidation state of its three cysteine residues [60]. HMGB1 within the nucleus is thought to be unmodified and in a reduced state. Upon stimulation for release, the NLS sites of HMGB1 become heavily acetylated, phosphorylated, or glycosylated, which promotes its translocation to the cytoplasm [26,48,61,62]. The release of HMGB1 from the cytoplasm into the extracellular milieu occurs through secretory lysosomes or autophagosomes, and is cell-type specific [27,48]. It will be interesting for future work to determine how the oxidation state of HMGB1 impacts the interaction with protein VII.

To investigate the relevance of the protein VII-HMGB1 interaction on chromatin, we focused on the type-I immune response through interferon. Our results show that in the presence of protein VII, expression of IFNβ is reduced to baseline levels. When HMGB1 is lost, this repression is partially rescued such that IFNβ levels were still induced by poly (dA:dT) treatment, though not to the same degree as in the control (Fig 7). We suspect that the presence of HMGB2 or HMGB3 in these cells may compensate for the lack of HMGB1, allowing protein VII to still repress IFNβ to a significant degree. This appears to be specific for IFNβ because transcription of downstream interferon stimulated genes was unchanged. These results indicate that protein VII's ability to repress IFNβ is not due to a general repression in host transcription, but a specific effect on IFNβ expression. HMGB1 and the remaining HMGB family of proteins are highly conserved across eukaryotes [63], while the interferon response is only found amongst vertebrate animals [64]. Because protein VII is well conserved in human serotypes [65,66], this suggests a potential mechanism by which protein VII from human adenoviruses has evolved to evade the innate immune system. Thus, hijacking HMGB1 by protein VII has a two-fold payoff: (1) it prevents HMGB1 release to dampen systemic inflammatory defenses and (2) it restricts IFNβ expression and in turn the downstream effects of interferon, both of which promote viral infection (Fig 6D).

There has been extensive work characterizing how many of the early adenovirus proteins inhibit the innate immune response to adenovirus infection. For example, E1A, E1B-55K, E4orf3, and E4orf6 all use strategies ranging from inhibiting transcription, and degrading host proteins, to sequestering host proteins to prevent the innate immune response [37,67–69]. Here, we propose a model in which protein VII hijacks HMGB1 to suppress IFNβ expression (Fig 7C). This is a unique strategy from the perspective of the virus since protein VII is a late viral protein and will likely not impact the death of the infected cell, but rather could prevent large amounts of interferon from being released upon cell death and lysis, thus dampening the systemic immune response. Furthermore, this appears to be separate from protein VII preventing the extracellular HMGB1-associated inflammation response.

In the past decade, several studies have reported that different viruses interact with and utilize HMGB1 for viral benefit. For example, the influenza virus nucleoprotein binds HMGB1 to promote replication [70] and SARS-CoV-2 infection also relies on HMGB1 to promote ACE2 expression [71]. In light of these studies, it will be fascinating to uncover the mechanisms by which HMGB1 functions not just in extracellular signaling as a response to infection but also as a target for hijacking within the nucleus to subvert host defenses.

## Materials and methods

### Material availability

All plasmids, primers, and cell lines used in this study are available upon request from the lead contact, D Avgousti.

### Bacterial growth conditions and chemical treatments

*Escherichia coli* strains were grown in LB (5 g / L NaCl, 10 g / L tryptone, 5 g / L yeast extract) at 37˚C with shaking at 200 rpm. Antibiotics were supplemented as needed (carbenicillin: 50 μg / mL liquid / 100 μg / mL plate, and kanamycin: 30 μg / mL / 50 μg / mL). Optical density was measured at 600 nm using an Eppendorf BioPhotometer. For bacterial two-hybrid assays, *E. coli* BTH101 was grown in M63 minimal media (0.4 g / L ammonium sulfate, 2.72 g / L monopotassium phosphate, 0.1 mg / L ferrous sulfate heptahydrate [pH 7.0], 1 mM magnesium sulfate, 0.2% maltose, 0.4% glucose, and 0.0001% vitamin B1) supplemented with

carbenicillin, kanamycin, and 1 mM IPTG for 48 hours at 30˚C with shaking before spotting on MacConkey agar (BD Difco) plates.

## Plasmid construction

HMGB1 plasmids for cell line construction were generated in a 2nd generation pLVX-M-puro transfer plasmid (Addgene plasmid# 125839). The HMGB1 sequence was obtained from the pcDNA3.1 Flag hHMGB1 plasmid (Addgene plasmid #31609). Different HMGB1 constructs (A-box, B-box, AB-box, BC-tail, GFP) were PCR amplified from the HMGB1 sequence, restriction enzyme digested and cloned into the pLVX transfer plasmid using traditional cloning methods. The mutHMGB1 with 6xHis tag was custom generated by GeneWiz, and then cloned into pLVX transfer plasmid using traditional cloning methods.

Constructs of HMGB1 and protein VII for B2H assays were PCR amplified from the HMGMB1.pLVX and protein VII plasmids using KOD DNA polymerase. PCR products were verified on a 1% TAE agarose gel and extracted using the Zymo Research DNA Clean and Concentrator Kit. PCR products were then digested using KpnI-HF and XbaI (New England Biolabs) and ligated into pUT18, pUT18C, pKT25, or pKNT25 digested with the same enzymes pre-treated with quick CIP (New England Biolabs). Ligated plasmids were then transformed into DH5α *E. coli* and validated by sanger sequencing.

## Bacterial two-hybrid analysis

Plasmids (pUT18, pUT18C, pKT25, and pKNT25) containing fusion constructs of HMGB1, protein VII, and variants were co-transformed into the B2H assay strain BTH101. Four replicates from each transformation were picked, grown in M63 minimal media for 48 hours, and spotted on MacConkey agar plates supplemented with carbenicillin, kanamycin, 1% maltose, and 1mM IPTG. Plates were incubated at 30˚C and imaged after 72 hrs. Interaction between protein VII and HMGB1, A-box, B-box, BC, and itself (VII dimerization experiments) was probed in four orientations (e.g., T25-VII with T18-HMGB1, T25-VII with HMGB1-T18, VII-T25 with T18 HMGB1, and VII-T25 with HMGB1-T18). Protein VII with AB and HMGB1 DNA-bending mutants was examined using C-terminal fusions (e.g., T25-VII with T18-AB). Similarly, pre-VII and VIIΔPTM with HMGB1 variants or self-interaction experiments were examined with C-terminal fusions. A-box, BC, and HMGB1 self-interactions, and A-box dimerization with BC or HMGB1 were assayed with N-terminal fusions and cross-oriented tags (e.g., BC-T25 with T18-BC and BC-T25 with BC-T18) to assess if dimerization occurred in the same orientation as the observed interaction between HMGB1 and protein VII. AB, B-box, DNA-bending mutant self-interactions, and B-box dimerization with A-box or AB was assayed using C-terminal fusions (e.g., T25-A$^{mut}$ with T18-A$^{mut}$).

## Cell culture, cell line generation and viral infections

A549 cells were purchased from ATCC and cultured using standard methods in Kaighn's modification of Ham's F-12 medium (F-12K) containing 100 U/ml of penicillin and 100 mg/ml of streptomycin and supplemented with 10% fetal bovine serum (FBS). Tetracycline negative FBS was used for cell lines containing dox-inducible protein VII constructs. HEK293T cells were purchased from ATCC and cultured using standard methods in Dubelco's Modified Eagle Medium (DMEM) containing 100 U/mL of penicillin and 100 mg/ml of streptomycin and supplemented with 10% fetal bovine serum (FBS) and used for generating lentivirus.

Lentiviruses were generated using 2nd generation pLVX transfer plasmid containing different constructs of HMGB1 (HMGB1.eGFP, Abox.eGFP, Bbox.eGFP, ABbox.eGFP, BCtail. eGFP, eGFP, and mutHMGB1.His). The pLVX, psPAX.2, and pMDG.2 plasmids were

transfected into a confluent 10cm$^2$ dish HEK293T cells at a ratio of 1: 3.75: 2.5, respectively. Supernatant was collected at 24-, 48-, and 72-hours post transduction and stored at -80˚C. A549ΔHMGB1 cells were seeded in a 6-well plate and then transduced with the different HMGB1 construct lentivirus. At 24 hours post transduction, puromycin was added with fresh media at 2 μg/mL to select for transduced cells. Successfully transduced cells were kept under puromycin selection while they were expanded to be frozen or used for experiments.

All infections were performed using standard protocols with a multiplicity of infection of 10 (Ad5), 100 (rAd VII-GFP), or 50 (rAd GFP). Wild-type Ad5 was obtained from ATCC and recombinant adenovirus vectors rAd VII-GFP and rAd GFP were a gift from D. Curiel [72]. Ad5 VII-HA was generated using bacterial recombineering with an E3-deleted Ad5 bacterial artificial chromosome (BAC) vector [73,74]. Briefly, an HA tag was added to the protein VII gene through PCR amplification, and then protein VII was introduced into the Ad5 genome through recombineering. Successful recombineering was verified through restriction digest and Sanger sequencing. The Ad5 VII-HA genome was linearized by PacI endonuclease digestion, transfected into 293 cells, and amplified over several passages. Following amplification, the virus was extracted from cell pellets by freeze thawing. All viruses were purified using two rounds of ultracentrifugation on a cesium chloride gradient and stored in 20% glycerol at -80˚C. Viral titers were determined by plaque assay.

## Immunofluorescence microscopy

Immunofluorescence microscopy was performed using standard methods as previously described [16]. Briefly, cells were seeded on poly-L-lysine coated coverslips in a 24-well plate. Cells were infected with Ad5, and coverslips were collected at 18 hours post infection and fixed with 4% paraformaldehyde. After fixation, cells were permeabilized with 0.5% Triton-X, and blocked with 3% BSA. Cells were incubated with primary antibodies for 1 hour, washed three times in PBS, incubated with secondary antibody and DAPI for 1 hour in the dark, and then washed three times in PBS. Coverslips were then mounted on slides with ProLong Gold Antifade Mountant (Thermo Fisher Scientific), and then allowed to dry overnight. High resolution confocal microscopy was performed with a Leica Stellaris Confocal Microscope using a 63x oil objective.

## Nuclear pre-extraction

Cells were prepared for immunofluorescence microscopy as stated above. Before fixation, cells were treated with nuclear pre-extraction buffer (20mM Hepes, 20mM NaCl, 5mM MgCl$_2$, 1mM DTT, 0.5% NP-40, 1X phosphatase inhibitor and 1X protease inhibitor) for 20 minutes on ice. After this treatment, cells were washed with PBS and then fixation with paraformaldehyde and the staining process proceeded as stated above.

## Fractionation and western blotting

The cellular fractionation protocol was adapted from Suzuki *et al.* [75]. Briefly, cell pellets were thawed on ice for 5 minutes and then resuspended in 900μL of ice-cold 0.5% NP-40. 300 μL was added to 100μL of 4X sample buffer as whole cell lysate. The remaining 600μL was incubated at room temperature for 5 minutes and then centrifuged at 13,000 RPM for 30 seconds. 300μL of the supernatant was added to 100μL of 4X sample buffer as the soluble fraction. The remaining 300μL was discarded and the pellet was resuspended in 900μL of ice-cold 0.5% NP-40 and incubated at room temperature for 5 minutes. The mixture was centrifuged at 13,000 RPM for 30 seconds, the supernatant was discarded, and the remaining pellet was resuspended in 200μL of 1X sample buffer as the insoluble fraction. All three fractions were boiled at 95˚C

for 20 minutes. Samples were run on 12% poly-acrylimide gels and then transferred to nitro-cellulose. The membranes were blocked in 5% milk in TBST, and then probed with primary antibodies at 4°C overnight. Blots were then probed with secondary antibodies for one hour, developed with Clarity Western ECL Substrate, and then imaged with a Biorad ChemiDoc MP Imaging System.

## Antibodies

Commercially available antibodies were purchased through Abcam (GFP [ab290], HMGB1 [18526], H3 [ab1791]), Sigma-Aldrich (Vinculin [V9131]) and Clontech (His [631212]). Protein VII antibodies were a generous gift from the Gerace and Wodrich labs [76]. DBP antibodies were a generous gift from the Levine lab [77]. Secondary antibodies used for immunoblotting were obtained from Jackson ImmunoResearch (115-035-003 and 111-035-045). Secondary antibodies for immunofluorescence microscopy were obtained from Thermo Fisher Scientific (A11008, A-11011, A-21245, A-11001, A32727, A-21236).

## Interferon and poly(dA:dT) treatment

For the interferon treatment, A549 VII-HA cells were treated with or without doxycycline for four days at 0.2 μg/mL, with fresh doxycycline added each day. On day three, cells were treated with or without recombinant human IFNβ at 1000 U/mL. Cell pellets were collected on day four for RNA extraction using the NEB Monarch Total RNA Miniprep Kit. One μg of RNA was converted into cDNA using iScript Reverse Transcription Supermix (Bio-Rad), and then cDNA was used for qPCR with iTaq Universal SYBR Green Supermix. For poly(dA:dT) treatment, cells were transduced with recombinant adenovirus vector rAd VII-GFP at an MOI of 100 or recombinant adenovirus vector rAd GFP at an MOI of 50 to achieve equal levels of GFP positive cells (greater than 90%). This is because the recombinant adenovirus vector rAd VII-GFP is less stable, thus a higher apparent MOI was required to achive the same levels of GFP positive cells. At 24 hours post transduction cells were treated with poly(dA:dT) at 1 μg/mL for 8 hours, and then cell pellets were collected for droplet digital PCR. RNA was extracted from the cell pellets and 1 ug of RNA was converted to cDNA as above, for droplet digital PCR (ddPCR). The cDNA was serial diluted and then mixed with a primer-probe mix for IFNβ and commercially available Eukaryotic 18S rRNA Endogenous Control (VIC/MGB probe, primer (4319413E). The ddPCR mixture consisted of 12.5 μL of of a 2X ddPCR Supermix for Probes no dUTP (Bio-Rad: 1863024), 1.25 μL of each primer-probe mix, and 10 μL of the diluted template cDNA. 20 μL of each reaction mixture was loaded onto a disposable Dg8 cartridge (Bio-Rad 1864008) with 70 μL of droplet generation oil (Bio-Rad 1863005) and oil droplets were generated (QX200 Droplet Generator). Generated droplets were transferred to a 96-well PCR plate (Bio-Rad 12001925), and PCR amplification was performed with a Bio-Rad C100 Touch Thermal Cycler under the following conditions: 95°C for 10 minutes, 40 cycles of 94°C for 30 seconds and 60°C for 1 minute with a ramp rate of 2°C/sec, followed by 98°C for 10 minutes, and then held at 12°C. Following amplification, the plates were loaded on a droplet reader (Bio-Rad QX200) and droplets were analyzed with droplet reader oil (Bio-Rad 1863004). Data was analyzed using the Quantasoft analysis software, and after accounting for fold-dilution of cDNA, the number of IFNβ positive droplets was normalized to every 1 million positive 18S rRNA positive droplets.

## Statistical analysis

All statistical analyses were performed using GraphPad Prism v9 or v10. Statistical tests and n values are described in the figure legends. Statistical significance was defined as $p<0.05\%$ in all

experiments. Specifically, we used a Welch's t-test and Student's t-test where described. Only p-values less than cutoff are reported in figures.

## Supporting information

**S1 Table. List of tested protein VII and HMGB1 bacterial two-hybrid interactions.** Each interaction was tested using four biological replicates. Legend with representative images for each phenotype is provided at the bottom. Positive interactions are indicated in green. (PDF)

**S1 Fig.** A. B2H assay of HMGB1 B-box in different orientations with protein VII. B. B2H assay between protein VII and indicated HMGB1 variants in three additional orientations. C. B2H assay of HMGB1 A-box with indicated HMGB1 variants showing self-interaction in one instance. D. B2H assay of protein VII N-terminal T25- and N-terminal T18-fusions (top) and an assay of protein VII mutants with alanine substitutions to PTM sites (VIIΔPTM) N-terminal T25- and N-terminal T18-fusions (bottom). Both assays show modest self-association. For results of interactions and controls in additional orientations, see S1 Table. (TIF)

**S2 Fig.** A. Immunofluorescence images showing all channels of mock infected cells from Fig 2C with HMGB1 in green, protein VII in magenta, and DAPI in gray (cyan in merge). Scale bar is 10 μm. B. Immunofluorescence images of Ad5 infected cells showing all channels from Fig 2C with HMGB1 in green, protein VII in magenta, and DAPI in gray (cyan in merge). Scale bar is 10 μm. (TIF)

**S3 Fig.** Western blots showing all controls that correspond with Fig 3C for protein VII, H3, GFP, and vinculin: HMGB1 (A), A-box (B), B-box (C), AB-box (D), BC-tail (E), and GFP (F). Western blots showing all controls that correspond with Fig 3E for protein VII, H3, GFP, and vinculin; HMGB1 (G), AB-box (H), and BC-tail (I). (TIF)

**S4 Fig. Western blot of DBP (early viral protein), adenoviral late proteins, and H3 loading control of infected WCL samples of cell lines from Fig 3C showing similar amounts of viral protein accumulation during infection of each cell line.** (TIF)

## Acknowledgments

We thank members of the Avgousti lab, Geballe lab, J Hyde, and J Smith for invaluable input and insightful comments. We thank former members of the Weitzman lab N Pancholi and C Herrmann for providing critical input. We thank D Nguyen and H Arbach for technical assistance. We thank M Valdez Cabrera for assistance with statistical analyses.

## Author Contributions

**Conceptualization:** Edward A. Arnold, Monica S. Guo, Daphne C. Avgousti.

**Data curation:** Edward A. Arnold, Robin J. Kaai, Katie Leung, Mia R. Brinkley.

**Formal analysis:** Edward A. Arnold, Katie Leung, Laurel E. Kelnhofer-Millevolte, Monica S. Guo, Daphne C. Avgousti.

**Funding acquisition:** Edward A. Arnold, Daphne C. Avgousti.

**Investigation:** Edward A. Arnold, Robin J. Kaai, Katie Leung, Mia R. Brinkley, Monica S. Guo.

**Methodology:** Edward A. Arnold, Katie Leung, Monica S. Guo, Daphne C. Avgousti.

**Project administration:** Monica S. Guo, Daphne C. Avgousti.

**Resources:** Edward A. Arnold, Robin J. Kaai, Katie Leung.

**Supervision:** Monica S. Guo, Daphne C. Avgousti.

**Validation:** Edward A. Arnold.

**Visualization:** Edward A. Arnold, Monica S. Guo, Daphne C. Avgousti.

**Writing – original draft:** Edward A. Arnold, Katie Leung, Monica S. Guo, Daphne C. Avgousti.

**Writing – review & editing:** Edward A. Arnold, Robin J. Kaai, Katie Leung, Mia R. Brinkley, Laurel E. Kelnhofer-Millevolte, Monica S. Guo, Daphne C. Avgousti.

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
