## [Decision Letter · Decision Letter 0]

11 May 2023

Dear Dr. Avgousti,

Thank you very much for submitting your manuscript "Adenovirus protein VII binds the A-box of HMGB1 to repress interferon responses." for consideration at PLOS Pathogens. As with all papers reviewed by the journal, your manuscript was reviewed by members of the editorial board and by several independent reviewers. In light of the reviews (below this email), we would like to invite the resubmission of a significantly-revised version that takes into account the reviewers' comments.

The reviewers are intrigued by your study but find areas of it deficient in quantitation, statistical analyses, and the implementation of controls.

We cannot make any decision about publication until we have seen the revised manuscript and your response to the reviewers' comments. Your revised manuscript is also likely to be sent to reviewers for further evaluation.

Sincerely,

Robert F. Kalejta

Academic Editor

PLOS Pathogens

Blossom Damania

Section Editor

PLOS Pathogens

Kasturi Haldar

Editor-in-Chief

PLOS Pathogens

orcid.org/0000-0001-5065-158X

Michael Malim

Editor-in-Chief

PLOS Pathogens

orcid.org/0000-0002-7699-2064

The reviewers are intrigued by your study but find areas of it deficient in quantitation, statistical analyses, and the implementation of controls.

Reviewer's Responses to Questions

**Part I - Summary**

Reviewer #1: Arnold et al report further studies characterizing the interaction of HAdV pVII with HMGB1. Bacterial 2-hybrid tests are done to demonstrate direct interactions and map interaction domains. A HMGB1 KO line is used for some interesting reconstitution studies looking at pVII effects on HMGB1 localization, again mapping domains and looking at the impact of mutations affecting known post-translational modification. This data includes a solubility assay that I thought was quite nice as surrogate for tight chromatin association. Ultimately, a convincing effect on repression of IFNb expression by pVII is demonstrated, which is an exciting new finding.

Reviewer #2: The authors previously demonstrated that adenovirus core protein VII, a histone-like viral protein, binds the high mobility group protein HMGB1 and sequesters this protein to chromatin. This process prevents HMGB1 from being released from cells and disrupts its ability to function as an alarmin to promote inflammatory signaling. In this submission, the authors characterize this process further. Using a bacterial one-hybrid approach followed by immunoflouresence microscopy in A549 cells, they demonstrate that protein VII binds to the A domain of HMGB1. Protein VII sequestration of HMGB1 results in ~75% of the HMGB1 pool fractionating in an insoluble fraction; the HMGB1 A box is necessary and sufficient for this to occur. HMGB1 DNA binding is not required. The authors previously showed that protein VII is post-translationally modified and this contributes to chromatin association. Here they show that VII PTMs are not required for HMGB1 binding but are required to confer insolubility. Finally, the authors show that protein VII expression does not inhibit an interferon-beta response, but does repress the induction of interferon-beta expression by poly(dAdT).

The manuscript is clearly written and overall the data are convincing. The manuscript does not address underlying mechanisms of activity (why VII PTMs are not required to relocalize HMGB1 but are required to induce HMGB1 insolubility, how VII regulates the induction of interferon-beta expression by polydAdT), thus the impact of the study is limited.

Reviewer #3: Previous research by Dr. Avgousti and associates has established a remarkable relationship between the major adenovirus "chromatin" protein, protein VII, and the cellular high-mobility group box 1 (HMGB1) "chromatin" protein. This relationship is of interest and is significant because protein VII has been shown to dampen the cellular anti-viral response. This has been shown to be due, at least in part, to the action of protein VII on HMGB1. This curious chromatin-associate protein is released by damaged cells where it acts as a chemokine or cytokine to signal danger. This makes understanding the nature of both of these part-time chromatin proteins important.

Using a rather comprehensive collection of constructs to study protein-protein interactions in bacteria, it was convincingly shown that one of the two DNA-binding domains in HMGB1 (the A-box) directly interacts with protein VII and that this can occur in the absence of posttranslational modifications known to affect protein VII function and in the absence of the DNA-binding property of HMGB1. Another intriguing, but inadequately supported observation is that this interaction likely results in the retention of HMGB1 in the nucleus of the infected cell. Although the interaction between HMGB1 and protein VII does not require posttranslational modifications to protein VII, these modifications may be important for the capacity of protein VII to render HMGB1 less freely soluble.

The strength of this report is that the findings in bacteria are clear and convinced and--if substantiated--the findings observed in the eukaryotic cell would be important. A significant limitation to this report is the failure to adequately quantify and validate the localization findings described in several figures.

The presentation, writing and clarity of the report is solid.

**Part II – Major Issues: Key Experiments Required for Acceptance**

Reviewer #1: none

Reviewer #2: The results in Figs. 6B/C are difficult to interpret. HMGB1 expression alone reduces the induction of interferon-beta expression ~5-fold (estimated by this reviewer from 6B) and protein VII alone reduces interferon-beta expression with control GFP 10-fold (quantified in 6C). The net change when both HMGB1 and VII are expressed is an additional 4-fold decrease in interferon-beta expression (6C). This is a statistically significant but rather modest effect. What is missing from this experiment is a control adenovirus vector to control for Ad-VII infection, ie. that the adenovirus vector infection does not contribute to the observed effect. Also, if HMBG1 and VII coordinate to inhibit the induction of interferon-beta expression by poly(dAdT), then one would anticipate that this should be found with other inducers of interferon-beta expression such as poly(IC) and RNA virus infection.

Reviewer #3: HMGB1, which is normally released from cells treated with the detergent NP-40, is retained in an insoluble form by protein VII through the A-box. It seems likely that this finding is significant but this was not appropriately demonstrated. First, the results in Fig. 2C are not convincing. If the difference being highlighted is differential localization within the nucleus, this should be systematically quantified and reported. The small sampling of examples in Fig. 2C display a rather broad diversity of localization patterns with insignificant numbers of examples to warrant any definitive conclusions. Similarly, the four and one-half cells shown in Fig. 5C inadequately support the conclusions. There are well-established means of selectively solubilizing cells while preserving the morphological integrity of the cell. Application of these methods together with more rigorous quantitative image analysis would likely address this shortcoming and provide valuable insight.

A second concern regards the use of the E1-deleted recombinant adenovirus to express protein VII. Replication-defective adenovirus vectors express more than just the transgene. The E4 gene products are expressed at non-trivial levels and E4orf1, E4orf3 and E4orf6 have been reported to have biological effects at levels nearly impossible to detect (see for example Ramalingam, 1999, https://doi.org/10.1182/blood.V93.9.2936). An appropriate control for the VII-expressing recombinant virus would be one that expresses an irrelevant gene. Was this done? If so, the results need to be reported. If not, this control needs to be added.

The third issue is that the statistical tests applied here were not adequately described. What specific tests in Prism were used? What was the correction for the multiple comparisons reported in several figures? If parametric tests were used for the statistical analysis, were the residuals appropriate for these tests? If so, report this.

**Part III – Minor Issues: Editorial and Data Presentation Modifications**

Reviewer #1: Minor comments relate to the data:

For figure 2 and any others with confocal microscopy, calculating overlap in signals for HMGB1 and pVII with a Pearson's correlation for statistical significance would be appropriate. Similarly, this can be done for fraction of HMGB1 at various subcellular locations, like the nucleolus.

Putting standard deviation significance indications on the bar graphs would be a good idea.

Minor textual suggestions:

I believe that all the HAdV reagents here are from HAdV-C5, but it wasn't specified anywhere that I noticed other than for describing the virus infection experiments.

Line 296 - rather than say "present in human cells", it might be better to say mammalian or eukaryotic cells.

Lines 350-351 - the HMGB1 independent effects of pVII on IFNb expression are very interesting. Is it possible that pVII can bind another HMGB protein (like HMGB2) in the absence of HMGB1 that has overlapping functions?

Line 460 - depletes sounds a little awkward here. Would limits or restricts be a better choice?

-The source of the A549 HMGB1 KO cells is not explicitly described in the methods.

-Some of the title of the methods sections have inconsistent internal capitalization compared to the others.

Reviewer #2: Line 55 - the literature (ref 9) shows that protein VII is released from the adenovirus gene during the early phase of infection.

Line 140 - more accurately state that protein VII is cleaved during the process of virion maturation.

The GFP images are shown in black and white - I think they would be easier to visualize in color and the merge images show GFP in color.

Fig. 4E - how is a value >1 obtained (ie. more HMBG1 is insoluble than present in the control lysate)?

Fig. 5B - I question if the results with the VII-PTM mutant in the bacterial one-hybrid assay should even be included in the manuscript. These modifications will not occur in bacteria with wild-type VII so what is the point?

Lines 321-322 - the title of this section is not accurate. Fig. 6A shows that protein VII does not inhibit the type-1 interferon response since no effect was observed on the induction of ISGs following interferon-beta addition. Rather, the data indicate that protein VII and HMGB1 inhibit interferon signaling.

Reviewer #3: The phrase "human cell biological systems" is rather devoid of meaning in the context of the abstract. A more precise term would be better.

While understandable, the term "preVII" has not been the term historically used for precursor adenovirus proteins (although it should have been.) It is helpful in the figures but rather than introduce yet another element of nomenclature, perhaps it should be kept as "pVII" in the text and a note made in the figure legends of this change.

Formally speaking, one does not "mutate" hydrophobic residues. It would more correct to simply say that these residues were changed (by a mutation in the gene.)

Figure legends generally fail to provide sufficient information to understand the figure. The nature of the error bars is not explained in the legends.

PLOS authors have the option to publish the peer review history of their article (what does this mean?). If published, this will include your full peer review and any attached files.

Reviewer #1: No

Reviewer #2: No

Reviewer #3: No
---

## [Decision Letter · Decision Letter 1]

17 Aug 2023

Dear Dr. Avgousti,

Thank you very much for submitting your manuscript "Adenovirus protein VII binds the A-box of HMGB1 to repress interferon responses." for consideration at PLOS Pathogens. As with all papers reviewed by the journal, your manuscript was reviewed by members of the editorial board and by several independent reviewers. The reviewers appreciated the attention to an important topic. Based on the reviews, we are likely to accept this manuscript for publication, providing that you modify the manuscript according to the review recommendations.

The reviewers were please with the manuscript revisions. There remains a few lingering issues, most notably with Figure 7B. It would appear that alterations to the text should be all that is required to mitigate these concerns. The authors are encouraged to address all of the comments form the reviewers. A final submission of the manuscript would be editorially evaluated rapidly.

Sincerely,

Robert F. Kalejta

Academic Editor

PLOS Pathogens

Blossom Damania

Section Editor

PLOS Pathogens

Kasturi Haldar

Editor-in-Chief

PLOS Pathogens

orcid.org/0000-0001-5065-158X

Michael Malim

Editor-in-Chief

PLOS Pathogens

orcid.org/0000-0002-7699-2064

The reviewers were please with the manuscript revisions. There remains a few lingering issues, most notably with Figure 7B. It would appear that alterations to the text should be all that is required to mitigate these concerns. The authors are encouraged to address all of the comments form the reviewers. A final submission of the manuscript would be editorially evaluated rapidly.

Reviewer Comments (if any, and for reference):

Reviewer's Responses to Questions

**Part I - Summary**

Reviewer #1: Arnold et al have resubmitted a repaired version of their continuing studies of the interaction of HAdV pVII with HMGB1. The list of minor corrections I provided have been addressed. I am happy with the improved quantification and the use of a better control for the pVII overexpressing virus. I don't think any additional mechanistic details have been teased out, but I was already happy with the extent of the data provided in the initial manuscript.

Reviewer #2: The authors previously demonstrated that adenovirus core protein VII, a histone-like viral protein, binds the high mobility group protein HMGB1 and sequesters this protein to chromatin. This process prevents HMGB1 from being released from cells and disrupts its ability to function as an alarmin to promote inflammatory signaling. In this submission, the authors characterize this process further. Using a bacterial one-hybrid approach followed by immunofluorescence microscopy in A549 cells, they demonstrate that protein VII binds to the A domain of HMGB1. Protein VII sequestration of HMGB1 results in ~75% of the HMGB1 pool fractionating in an insoluble fraction; the HMGB1 A box is necessary and sufficient for this to occur. HMGB1 DNA binding is not required. The authors previously showed that protein VII is post-translationally modified and this contributes to chromatin association. Here they show that VII PTMs are not required for HMGB1 binding but are required to confer insolubility. Finally, the authors show that protein VII expression does not inhibit an interferon-beta response, but does repress the induction of interferon-beta expression by poly(dAdT).

The manuscript is clearly written and overall the data are convincing. The manuscript does not address underlying mechanisms of activity (why VII PTMs are not required to relocalize HMGB1 but are required to induce HMGB1 insolubility, how VII regulates the induction of interferon-beta expression by polydAdT), thus the impact of the study is limited.

Reviewer #3: The strength of this report is the novel methods used to study the biochemical properties of a key adenovirus chromatin protein and cellular regulator. Previous weaknesses have been rectified and appropriate statistical and quantitative measures are provided that justify the conclusions.

**Part II – Major Issues: Key Experiments Required for Acceptance**

Reviewer #1: none

Reviewer #2: (No Response)

Reviewer #3: None required.

**Part III – Minor Issues: Editorial and Data Presentation Modifications**

Reviewer #1: 1) Line 188: Generally the word serotype is not used with regards to adenovirus anymore. This should just be "type", or more commonly HAdV-C5

2) Line 341: Not sure about the jump to cell cycle here. Seems like a big speculative leap without any basis.

3) Line 382: The sentence starting with "Strikingly" is a bit confusing placed where it is. The strikingly would be better moved to describe the complete abrogation of basal INFbeta expression by pVII in the deltaHMGB1 cells not treated with poly(dA:dT). I found that striking.

4) Some of the journal titles in the refs seem a little off the normal abbreviations. What's with the dagger symbol in ref 44 line 833?

Reviewer #2: Fig. 7 is improved with the required Ad-GFP control virus, but there are two aspects in 7B that require clarification. First, the authors state in lines 384-387: "In the absence of HMGB1 (A549ΔHMGB1 cells), we found a robust increase in IFNβ expression upon stimulation with poly(dA:dT) that reached significantly higher levels than those of the GFP expressing A549 cells." The figure shows that the statistical significance is higher (P = <.05 vs p = <.005) but the magnitude of the difference between the two cell lines does not appear to be that great. If the authors want to make this claim, they need to state the numerical differences between the Ad-GFP +/- poly(dA:dT) in the A549 vs. A549-dlHMGB1 cells. Second, the A549-dlHMGB1 cells + Ad-VII-GFP, No Treatment, sample (7th bar) shows a complete loss of IFNbeta expression compared to the Ad-GFP control, and compared to the two No Treatment samples in A549 cells (left side). There is no explanation given for this result. What is the level of IFNbeta expression in uninfected A549 cells? Do the levels seen in Ad-GFP and Ad-VII-GFP infected cells, No Treatment, represent an induction of IFNbeta expression compared to control, uninfected cells? Also, given the greater than 5 log difference in Ad-VII-GFP infected A549dlHMGB1 with No Treatment vs. poly(dA:dT), how can the significance only be <.05? Something is amiss here and it certainly requires an explanation and/or modification.

Reviewer #3: None identified.

PLOS authors have the option to publish the peer review history of their article (what does this mean?). If published, this will include your full peer review and any attached files.

Reviewer #1: No

Reviewer #2: No

Reviewer #3: No

Figure Files:

Data Requirements:

Reproducibility:

References:

---

## [Editor Report · Decision Letter 2]

23 Aug 2023

Dear Dr. Avgousti,

We are pleased to inform you that your manuscript 'Adenovirus protein VII binds the A-box of HMGB1 to repress interferon responses.' has been provisionally accepted for publication in PLOS Pathogens.

Best regards,

Robert F. Kalejta

Academic Editor

PLOS Pathogens

Blossom Damania

Section Editor

PLOS Pathogens

Kasturi Haldar

Editor-in-Chief

PLOS Pathogens

orcid.org/0000-0001-5065-158X

Michael Malim

Editor-in-Chief

PLOS Pathogens

orcid.org/0000-0002-7699-2064
---

## [Editor Report · Acceptance letter]

8 Sep 2023

Dear Dr. Avgousti,

We are delighted to inform you that your manuscript, "Adenovirus protein VII binds the A-box of HMGB1 to repress interferon responses.," has been formally accepted for publication in PLOS Pathogens.

Best regards,

Kasturi Haldar

Editor-in-Chief

PLOS Pathogens

orcid.org/0000-0001-5065-158X

Michael Malim

Editor-in-Chief

PLOS Pathogens

orcid.org/0000-0002-7699-2064